# A versatile marine modelling tool applied to arctic, temperate and tropical waters

**Janus Larsen** [ID]*, **Christian Mohn**[☉], **Ane Pastor** [ID][☉], **Marie Maar**[☉]

Department of Bioscience, Aarhus University, Roskilde, Denmark

☉ These authors contributed equally to this work.
* janus@bios.au.dk

**Data Availability Statement:** At the FlexSem webSite: http://marweb.dmu.dk/Flexsem At the CMEMS website: http://marine.copernicus.eu/ At the nsidc website: https://nsidc.org/data/IDBMG4 At the C3S website:

## Abstract

The improved understanding of complex interactions of marine ecosystem components makes the use of fully coupled hydrodynamic, biogeochemical and individual based models more and more relevant. At the same time, the increasing complexity of the models and diverse user backgrounds calls for improved user friendliness and flexibility of the model systems. We present FlexSem, a versatile and user-friendly framework for 3D hydrodynamic, biogeochemical, individual based and sediment transport modelling. The purpose of the framework is to enable natural scientists to conduct advanced 3D simulations in the marine environment, including any relevant processes. This is made possible by providing a precompiled portable framework, which still enables the user to pick any combination of models and provide user defined equation systems to be solved during the simulation. We here present the ideas behind the framework design, the implementation and documentation of the numerical solution to the Navier-Stokes equations in the hydrodynamic module, the surface heat budget model, the pelagic and benthic equation solvers and the Lagrangian movement of the agents in the agent based model. Five examples of different applications of the system are shown: 1) Hydrodynamics in the Disko Bay in west Greenland, 2) A biogeochemical pelagic and benthic model in the inner Danish waters, 3) A generic mussel farm model featuring offline physics, food levels and mussel eco-physiology, 4) Sediment transport in Clarion-Clipperton zone at the bottom of the Pacific and 5) Hydrodynamics coupled with an agent based model around Zanzibar in Tanzania. Hence we demonstrate that the model can be set up for any area with enough forcing data and used to solve a wide range of applications.

## Introduction

Holistic modelling of marine ecosystems is an increasingly essential and recommended approach to support integrated assessments of vulnerable marine ecosystems and implementation of adaptive marine management strategies and directives [1, 2]. Improved understanding of linkages and drivers across different ecosystem components over the past decades, in combination with technical advances of computing resources and infrastructures, have created a wealth of ecosystem models and modelling frameworks in marine science. Design philosophy

https://cds.climate.copernicus.eu At the ODA website: https://odaforalle.au.dk At the GEMCO website: https://www.gebco.net/ At the OCETide website: http://people.oregonstate.edu/~erofeevs/Afr.html.

**Funding:** This work was supported by Blue Nodules (EU Horizon 2020 Grant 688975), TASSEEF (Danish AgriFish Agency, grant no. 33113-I-16-011), INTAROS (EU Horizon 2020 Grant 727890), BONUS OPTIMUS (the EU and the Innovation Fund Denmark), and the Baltic nest institute.

**Competing interests:** The authors have declared that no competing interests exist.

and complexity of available models represent a wide range of applications and problems including simplified box models [3–5], spatially explicit end-to-end models [6–9], and complex 3D high-resolution models [10–13]. The predictive and analytical skill of ecosystem models is largely determined by their proficiency to resolve the dynamical scales of key processes, drivers and feedbacks [14, 15]. To achieve this, modern 3D ocean models apply numerical techniques to increase horizontal and vertical resolution in areas of specific interest. Flexible grid refinement in the horizontal is provided by unstructured meshes and by structured meshes through nesting or orthogonal curvilinear coordinates. In the vertical, different choices of traditional (e.g. z-level, terrain-following (sigma), isopycnic) and hybrid coordinates are available. Table 1 provides a non-exhaustive list of popular and widely distributed ocean model systems with representations of different marine ecosystem components, their main characteristics and key features. Ocean models adopting structured meshes and nesting capabilities include the ROMS/Croco model family (e.g. [16]), NEMO-Top/Pisces (e.g. [17]) and MITgcm [18]. Improved numerics of unstructured-mesh techniques now allow users to choose from a variety of different model codes with flexible grid geometry, but without additional nesting, including FVCOM [19], Delft3D-Delwaq [20] and MIKE3d-Ecolab [21, 22]. A third class of modelling tools exchange data and information between different model codes and model components. Popular examples of such coupling tools are the OASIS coupler [23] and the coupling framework FABM [24].

The majority of open-source community ocean models require a high level of programming skills and familiarization with the runtime environment (mainly Fortran codes on Unix operating systems) [25]. Other modelling software are either commercial, expensive, not open source and therefore often not suited for the use in academic research. The objective of the FlexSem model framework is to make advanced 3D unstructured marine modelling more available for a broader range of applications and users. The framework combines full ecosystem model capabilities with ease of use, thus offering non-specialists and early stage researchers easier access to ecosystem modelling without the need of expert knowledge in computer programming. A central design philosophy is to separate the programming of the numerical kernel of the model code from user applications and development of sub-models. Consequently, a very broad range of marine sub-models can be set up by the user by modifying xml-formatted text input files including all examples of use that we present in this paper, without the need for programming or compilation of model code. In this paper, we describe in detail the FlexSem modelling framework including all major components and sub-models. We further present a collection of model applications from different finished and ongoing research activities.

## Framework description

The framework is built around a 2D unstructured computational mesh. The mesh is made of nodes, with x and y coordinates and a connectivity which connects the nodes into polygons. Any combination of polygons can be used, but the mesh must be orthogonal, i.e. in every polygon there is a point, which when connected, the connection lines are perpendicular to the polygon interfaces. These points known as Voronoi points [26] are shown in Fig 1. Such tessellations can efficiently be created for complex topographies by mesh generators such as JigSaw [27].

Whereas both the pelagic and sediment models utilizes the same horizontal discretization, the vertical discretization can be freely defined at fixed depths (z-layer) in each model. The combination of an unstructured horizontal mesh and the vertical discretization allows for both 0D, 1D, 2D and 3D models. 0D is one computational cell only. 1D and 2D model can be either vertical (water column) or horizontal (area) and thus a very broad range of models from simple box models to high resolution 3D models can be implemented.

**Table 1. Non-exhaustive list of ocean model systems and their main characteristics and features.**

| Model system | Main characteristics | Ecosystem components | Runtime environment | Where to find |
|---|---|---|---|---|
| ROMS | Structured horizontal grid with nesting, terrain-following stretched vertical coordinates | Physics, biogeochemical, sediments | Unix, Fortran | www.myroms.org |
| Roms-Agrif / Croco | Structured horizontal grid with nesting, terrain-following stretched vertical coordinates | Physics, biogeochemical, sediments | Unix,Fortran | http://www.croco-ocean.org/ |
| NEMO-TOP/ PISCES | Structured horizontal grid with nesting, terrain-following stretched vertical coordinates | Physics, biogeochemical | Unix, Fortran | https://www.nemo-ocean.eu/ |
| MITgcm | Structured horizontal grid with nesting, z-coordinates | Physics, biogeochemical | Unix, Fortran | http://mitgcm.org |
| Atlantis | Polygon mesh in the horizontal | Aggregated physics from hydrodynamic models, end-2-end food web model | Windows, Mac, Unix, C/C++ | https://research.csiro.au/atlantis/ |
| MIKE Eco-lab | Unstructured horizontal grid, vertical?? | Physics, complete modelling framework for water quality and ecological modelling | Windows, Unix, C/C++ | https://www.mikepoweredbydhi.com/products/eco-lab |
| Delft3D Delwaq | Unstructured horizontal grid, vertical?? | Physics, water quality and ecological model | ?? | http://oss.deltares.nl/web/delft3d/delwaq |
| FVCOM | Unstructured finite volume horizontal grid, various topographic-following vertical coordinates. | Physics, biogeochemical, sediments, water quality | Unix, Fortran | http://fvcom.smast.umassd.edu/fvcom/ (Tian et al. 2015) |
| FABM | Structured grids | Coupling suite that connects a hydrodynamic model with multiple biogeochemical submodels. | Unix, Fortran | https://github.com/fabm-model/fabm/wiki |
| OASIS3-MCT | Supports structured and unstructured grids | Coupling tool that connects different components of the climate system | Unix, Fortran | https://portal.enes.org/oasis |

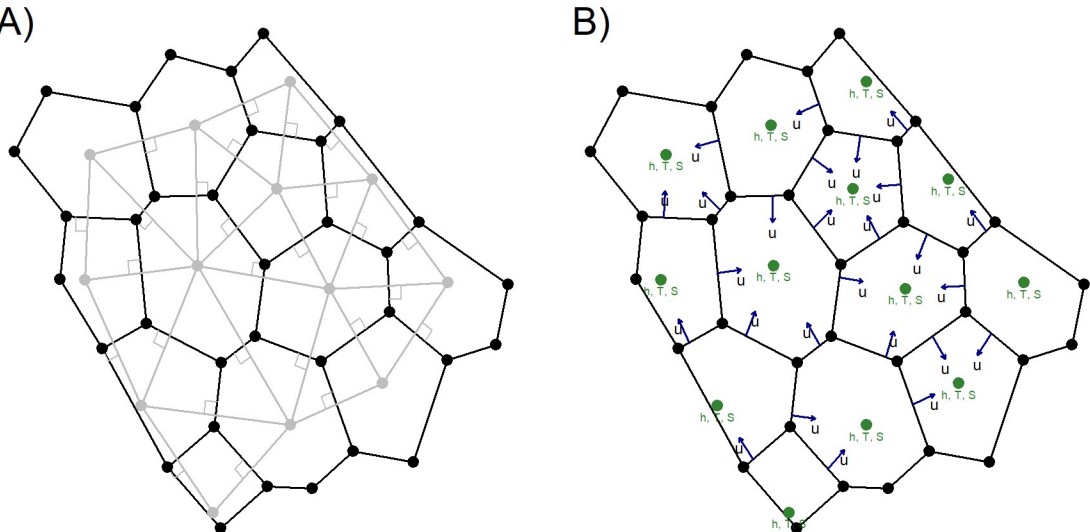

**Fig 1. Computational mesh.** A) An example of a 2D unstructured orthogonal mesh. Nodes in black, Voronoi points and perpendicular connecting lines (dual mesh) in grey. B) The staggering (C-grid) of the mesh: scalars are defined at the Voronoi points in the element/cell centers (finitie volume) and velocities on the dual mesh intersections points (finite difference). Velocities can be both positive and negative, thus representing any flux through the mesh.

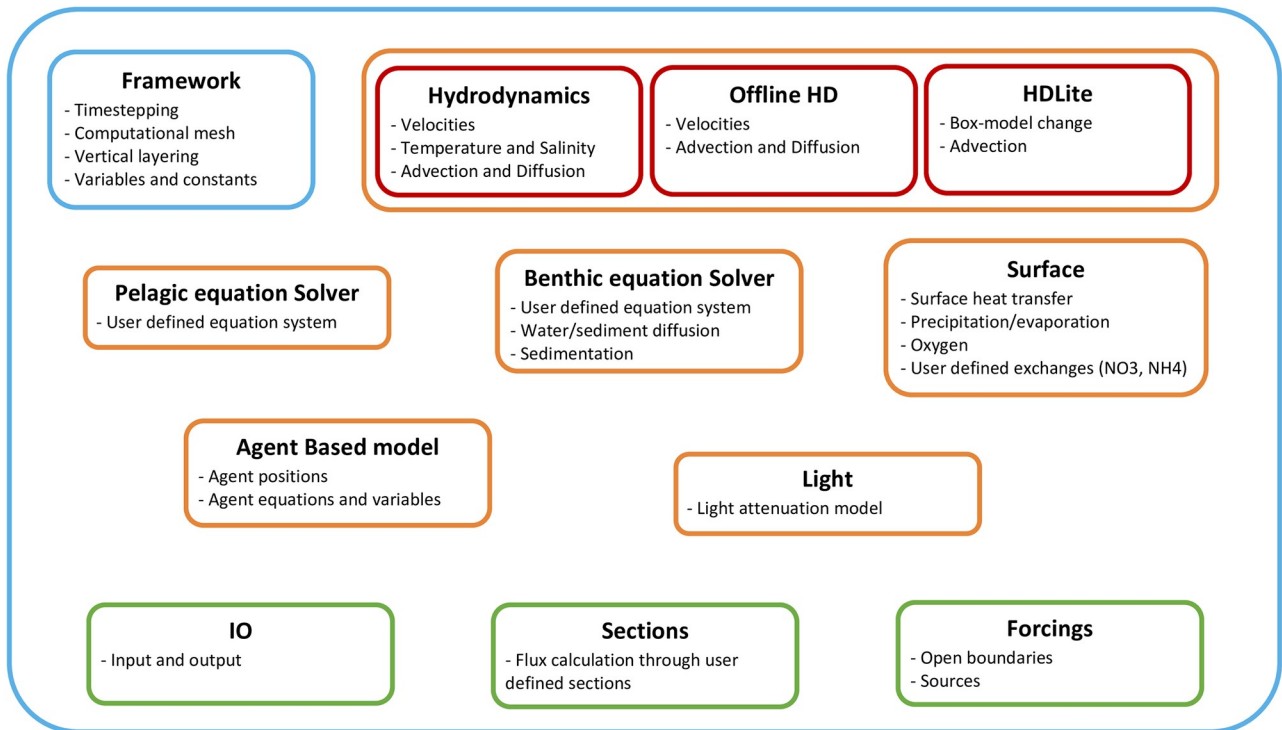

**Fig 2. Framework modules.** All modules (orange) in the framework can run independently or fully coupled, except the three hydrodynamic (HD) modules (red), which are alternatives. The only mandatory part is the framework it self (blue), which defines the computational mesh, simulation period, overall timestepping, variables and constants. Additional functionality are Input/output, sections and forcings (green).

Currently, modules for hydrodynamics, offline hydrodynamics, simplified hydrodynamics (HDLite), pelagic biogeochemistry and/or sediment transport model, benthic biogeochemistry, surface heat model, surface exchanges (e.g. oxygen, nutrients, precipitation), light penetration and agent based modelling (ABM) are implemented (Fig 2). The three hydrodynamic modules are alternatives, but all other modules can run independently or fully coupled in any combination. I.e. it is possible to run a biogeochemical pelagic model without hydrodynamics or an agent based model with offline or no hydrodynamics. This enables the user to create and test model parts, before putting it together into one model.

For learning and training purposes, ultra-simple and very fast box models with very simple equation system can, with very little instruction, be created and used in a few hours. This has been demonstrated during the Ph.D course "Mussel mitigation cultures; mussel growth, environmental quality and site selections" held 18–25 August 2019 at the Rønbjerg field station in Denmark where students with different backgrounds made their own models in an afternoon session. See also example in Appendix 1.

Hydrodynamics can either be calculated by the full hydrodynamic module—see detailed description below, calculated by simplified hydrodynamics [28] or prescribed as offline hydrodynamics. In the latter case, the information needed to advect state variables is read from a file. The offline module interpolates the offline data in time to match the time-stepping of the model. Offline files can be generated either by outputting it from a full hydrodynamic model run, by interpolating fluxes from a third party model or from measured velocity profiles [29].

Open boundaries can be defined by marking the relevant nodes in the mesh file and open boundary conditions can be applied to the pelagic model. The model implements clamped boundary conditions, i.e. when using the hydrodynamic module the user must specify

velocities and free surface height on the boundary and for the pelagic module boundary values must be specified for all advected state variables. Sources that add freshwater and modify other state variables such as nutrients can be defined anywhere in the computational mesh. Additional options for distributing source input in the water columns and creating dynamic sources that move around in time are also available.

For input and output, the framework provides support for text (fixed width or separated) and binary (native or NetCDF) formats. The text formats are convenient for time-series, profile and test data, whereas a binary format is usually used for 2D and 3D data. A R (https://www.r-project.org/) package is available for interfacing with the binary files.

The framework is open source, implemented in standard c++ and parallelized with openMP. The source code and binaries compiled for 64 bit windows as well as documentation are available under the GNU General Public License, Version 3 at the web site https://marweb.bios.au.dk/Flexsem/. The binary consists of a single executable file only and is therefore fully portable, i.e. no installation is needed and the program can be run after simply copying it to the computer. However, NetCdf 4 is a prerequisite and older windows versions may need to install the C++ redistributable from Microsoft.

## Hydrodynamics

Hydrodynamic modelling simulates the motions of the ocean, by solving a set of equations that gives the relationship between the fluid motion, i.e. the 3 velocity components, and the forces that act upon the fluid, e.g. gravitational force, pressure forces, wind and bottom drag and the Coriolis force. The Navier-Stokes equations accurately describes this relationship, but because an analytical solution cannot be found for real ocean cases, the equations are discretized in time and space and solved using numerical methods.

The hydrodynamic (HD) module implements a 3D semi-implicit, finite difference-finite volume, hydrostatic and nonhydrostatic solution to the Navier-Stokes equations on an unstructured computational mesh as outlined by [26]. Horizontal and vertical advection and diffusion of momentum is discretized using an eulerian second order Adam-Bashford approach following [30].

An orthogonal unstructured computational mesh is required for this discretization to be accurate. The velocities in the model are defined on the interfaces between the cells and the scalars in the cell centers, hence it is a finite difference-finite volume formulation known as the C-grid staggering [26, 31] (Fig 1).

To solve the Navier-Stokes equations, a predictor-corrector algorithm is used, where first, a provisional solution is calculated where the non-hydrostatic pressure component is neglected, thereby obtaining the hydrostatic (barotropic) solution. Subsequently the provisional solution can optionally be corrected by solving the implicit equation system for the non-hydrostatic pressure component, thereby obtaining the non-hydrostatic solution. Both the non-hydrostatic and hydrostatic solutions are mass conservative [26].

The viscosity in the model consist of a laminar term, which can freely be defined by the user and a dynamic term. The dynamic term can optionally be calculated using the Smagorinsky model which calculates the eddy viscosity as function of velocity shear [32]. Drag on bottom, surface and vertical walls can be defined by the user.

The chosen discretized formulation of the Navier-Stokes equations are semi-implicit, which ensures a both stable and accurate solution. A theta formulation is used to define the degree of implicitness. When theta equals zero, the model is explicit and when theta equals one it is fully implicit. It can be shown that the highest accuracy and efficiency is achieved when theta equals 0.5 [33].

Options to advect pelagic variables, such as temperature, salinity and biogeochemical state variables with either an explicit or a semi-implicit advection scheme are included in the framework. The latter is implicit in the vertical and explicit in the horizontal [26]. The advection module can also calculate fluxes through user defined sections in the model domain.

A surface radiation model simulates the heat transfer through the ocean surface. The heat fluxes are classified in radiation, convection, conduction, and evaporation. Whereas evaporation, conduction, and long wave radiation are surface effects, short wave radiation can penetrate into the water column [34].

The surface heat budget can in the framework be determined by 1) Fick's law of conduction, with all heat transferred to the surface layer or 2) Prescribed or calculated fluxes of short wave radiation, long wave radiation, sensible and latent heat.

If the fluxes are chosen to be calculated, the sensible and latent heat flux and the long wave radiation are calculated from the wind speed, cloudiness, water and air temperature as well as the relative humidity or dew point temperature. The absorbed short wave radiation is calculated from the clear sky incoming radiation, the cloudiness and the short wave surface reflectivity [34]. The clear sky incoming radiation is calculated from the solar constant, latitude and the day of year [35]. The sum of the heat fluxes from long wave radiation, sensible and latent heat is absorbed exponentially decreasing by depth until the penetration depth is reached. If the water is shallower than the penetration depth, the heat is reflected in the bottom and the remaining heat equally distributed in the water column [34].

The incoming short wave radiation has surface reflectivity, exponential extinction in the water column and reflection at the bottom. Like the other heat fluxes, the short wave radiation heat flux is also absorbed exponentially decreasing by depth until the penetration depth (~the Secchi disk depth) is reached. If the bottom is reached, the remaining flux is refracted and absorbed exponentially decreasing upwards toward the surface. If the surface is reached before all the heat is absorbed, the remaining is equally distributed in the water column.

## 3D pelagic and 3D benthic equation solver

In dynamic biogeochemical modelling, the biological, geological and chemical processes that govern the ocean biology are described with a set of differential equations, which when solved, simulates the state of the system. Variables in the pelagic, which are transported by the ocean currents, e.g. nutrients or phytoplankton biomass, are known as state variables. The state variables are transported around the model domain by solving the advection-diffusion equation for each state variable.

The framework has no build-in equation system for biogeochemical processes, but allows the user to define any custom equation systems e.g. one for pelagic and benthic biogeochemical processes. The user can define constants, variables and equations for both the pelagic and the benthic sub models. The equations are inputted via the xml formatted setup file, parsed by the framework and numerically solved in every time-step [29]. If a hydrodynamic model is included in the setup, the pelagic variables can optionally be transported by the chosen advection-diffusion scheme. All models are 2-way coupled and on the interface between the pelagic and benthic models, variables can be exchanged by diffusion. In addition, vertical settling velocities can be defined for both pelagic and benthic variables, where the pelagic variables will settle into the top layer of a given benthic variable. A number of other options are available for creating e.g. bottom stress induced resuspension, bathymetry correlated patchiness and freely defined movement of material.

The pelagic and benthic equation solver can also be used to implement a sediment transport model as has been done in the usage example "Sediment transport in the Clarion-Clipperton

Zone" outlined further below. Using the same syntax as for a biogeochemical model, the user can define different sediment grain size fractions and implement static or dynamic settling velocities and resupension of settled material as functions of e.g. bottom stress and grain size. Modification of the grain size distributions e.g. by flocculation can also readily be implemented. Coupling such a setup with a hydrodynamic module will allow for 3D transport of the suspended sediment.

This generic equation solver approach has proven highly versatile and been used to setup NPZD and microplankton-detritus model (MPD) type biogeochemical models e.g. in combination with dynamic energy budget (DEB) for blue mussels in suspended farms, models for fish aquaculture sites and for studying pelagic food web dynamics [36, 37]. It has also been used for sediment transport models in areas as different as the shallow Limfjord in northern Jutland, Denmark and the Clarion-Clipperon Zone in the deep pacific. The usage examples given further below also demonstrates the wide range of application.

## Agent based model

In Agent Based Modelling (ABM), traits of individuals, groups of individual or particles and their interaction with the surrounding environment are described by a set of differential equations. Ocean motion velocities are interpolated to the positions of each agent, which are transported using a Lagrangian approach. This enables the study of patterns of behavior from the description of individual traits. A classic application of ABM is to map habitat connectivity where the connection between a number of habitats are mapped by releasing agents in habitat and track were they end up, as it is done in the "Zanzibar ABM" usage example below.

The same equation solver as used for the pelagic and benthic equations can be used for the agents in the ABM module of the framework. This allows the user to freely define traits or environmental dependent behaviors of the agents. The agents can be activated or deactivated by user defined criteria. If 3D pelagic velocities are available in the simulation, typically from the hydrodynamic module, the agents can be transported around the model domain by Lagrangian transport. In this case, the velocities are interpolated to the position of each agent using an area based interpolation approach [38]. Movement of the agents can be modified by adding diffusivity, which is implemented as a uniform directional random component to the agent velocities and can be specified freely by the user as a constant or to depend on agent properties or environmental parameters.

The time stepping of the ABM module can be specified to any multiple of the overall time step. In each ABM time step the agents equations system is solved and the agent's positions are updated by advection and diffusion. If an agent is transported across a solid boundary in a given time-step, the agent is returned to the center of the boundary element maintaining the depth. Open boundaries can be set to act like solid boundaries or as sinks of agents, so that agents crossing an open boundary are deactivated.

## Usage examples

To demonstrate some possible applications of the FlexSem framework, we here include five examples of use. The examples are chosen to demonstrate the main modules of the framework and adaptation to different areas, but many more applications are possible and more examples of use can be found on the website https://marweb.bios.au.dk/Flexsem/.

## Disko Bay hydrodynamics

The Disko Bay is the largest bay on the west coast of Greenland. Connected to the Baffin Bay towards the west by a trench and a 300 m deep sill, it has an average depth of 400 m and

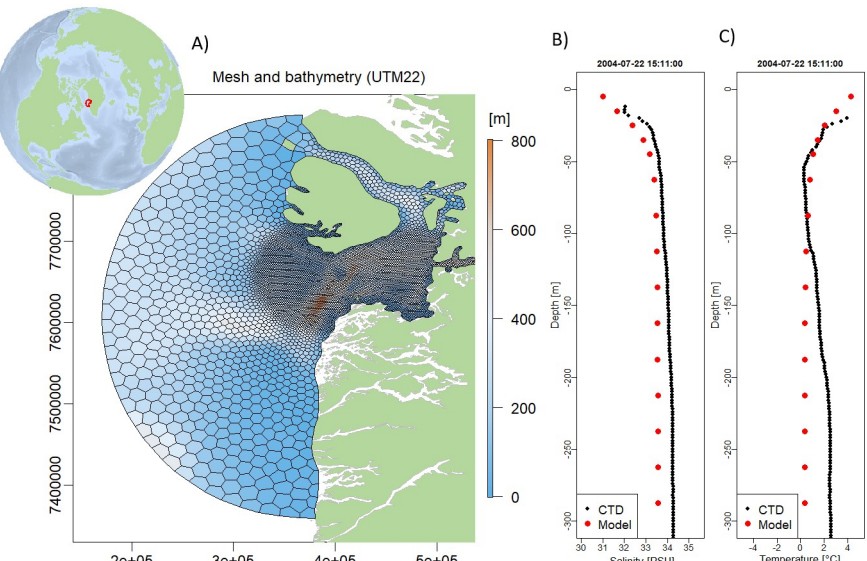

**Fig 3. Disko Bay in Western Greenland.** A) Location and computational mesh and B-C) Comparison of model and CTD measurements from 22 July 2004.

features a complex bathymetry of sills and trenches [39]. It has several long narrow sub fjords, among these the Ilulissat ice fjord with the world most productive glacier draining approximately 5.4% of the Greenland Ice Sheet [40]. Just south of the fjord, a shallow area up to only 30 m deep is Store Hellefisk Banke, an area of very high productivity and known rich fishing grounds. As part of the EU Horizon 2020 Integrated Arctic Observation System (INTAROS) project, a high resolution model was set up for the Disko Bay area, with the main goal of modelling the primary production. Here we present the hydrodynamic part of the model.

An orthogonal mesh centered at (68.59345N, 55.53959E) and varying in horizontal resolution from 2 km in the inner part to 15 km near the open boundary was constructed using Jig-Saw. The bathymetry was interpolated from the 150x150 m resolved IceBridge BedMachine Greenland, Version 3" bathymetry (https://nsidc.org/data/IDBMG4), see Fig 3A. The model has 25 vertical layers, increasing in thickness from 10 m in surface to 50 m near the bottom. Open boundary forcings of velocities, water level, temperature and salinity was obtained from the HYCOM model provided by the Danish Meteorological Institute (DMI). These data can also be downloaded from the Copernicus Marine Environment Monitoring Service (CMEMS). 2D fields of meteorological data was also provided by DMI and used for free surface forcings, but meteorological forcings can also be downloaded from the Copernicus Climate Change Service (C3S) at https://cds.climate.copernicus.eu. Meltwater run-off from Programme for Monitoring the Greenland Ice Sheet (PROMICE) was provided by the Geological Survey of Denmark and Greenland (GEUS) and used for freshwater input [41]. The time-step in the model was 5 minutes and the hydrostatic hydrodynamics including advection and diffusion of salinity and temperature simulates 1 year in 2.5 hours on an off-the-selves desktop pc (4 core intel i7-6400, 3408 MHz). More information about this setup can be found at the FlexSem website.

The model reproduces the tidal currents in the straits and the pumping of deeper water up on Store Hellefisk Banke. The modelled salinity and temperature profiles are compared with CTD measurements obtained from the International Council for the Exploration of the Seas (ICES) database (Fig 3B and 3C). The profiles are from the inner part of the Disko Bay near

the outlet of the Ilulissat ice fjord, and it can be seen that the model reproduces the fresh surface water and the general salinity profile, although the deeper water is slightly too fresh.

## Biogeochemical model—Horsens fjord area

The Horsens Fjord area is located in the Danish transition area between the North Sea and the Baltic Sea (Fig 4A). The area is relatively shallow (<36 m deep) and shows a high variability in salinity and nutrient concentrations due to the inflow from the brackish Baltic Sea and the high-salinity North Sea and inputs from local streams [42]. The ecological status is characterized as 'bad' to 'poor' according to the Water Framework Directive with high Chlorophyll a (Chl-a) concentrations and summer hypoxia in deeper waters (HELCOM 2017). The purpose of the coupled hydrodynamic-biogeochemical model was to estimate the environmental effects from mussel mitigation cultures as part of the BONUS OPTIMUS project.

The computational mesh centered at (55.7414N,10.2543W) consisted of hexagons varying from 130 to 1800 m in horizontal resolution with the smallest grid cells close to the mussel farm (Fig 4A). The vertical discretization was lowest with 1 m in the upper 4 m of the water column and highest with 10 m in the deepest layer (26-36m). At the open boundaries the hydrodynamic model was forced with water level, velocities, temperature and salinity data provided by CMEMS. Initial fields of salinity and temperature was also interpolated from

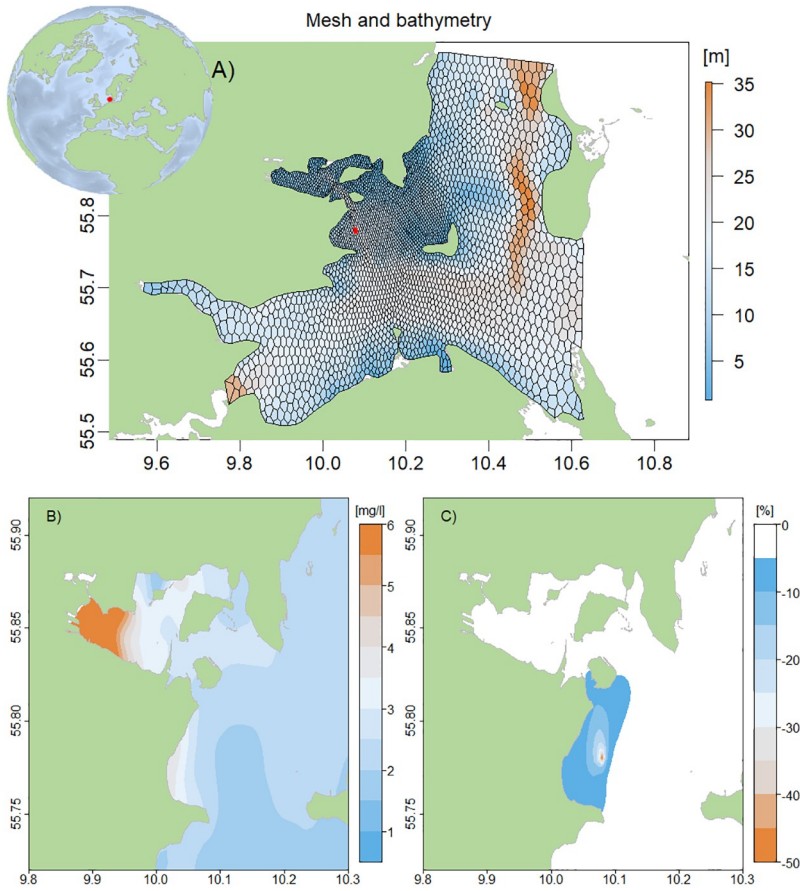

**Fig 4. Biogeochemical model—Horsens Fjord area.** A) Bathymetry and computational mesh. The red polygon indicates the location of the mussel farm B) average summer Chl a concentrations (mg/l) and C) %-change in Chl-a concentration around the mussel farm.

CMEMS data. The FlexSem hydrodynamics was coupled to the biogeochemical ERGOM model [36] applied in combination with a mussel farm model. ERGOM describes the nutrient cycling through uptake by three groups of phytoplankton, grazing by two zooplankton groups, production and remineralization of detritus as well as oxygen dynamics. Initial and forcing data for the ecological model was interpolated from data downloaded from the Danish National Monitoring Database (ODA) at https://odaforalle.au.dk. The mussel growth was described by a dynamic energy budget model in response to salinity, temperature and food [43] with feedbacks to the biogeochemistry through filtration of phytoplankton biomass and detritus, excretion of nutrients, respiration and biodeposits [44, 45]. The number of mussels in the farm decreased over time as the mussels grew larger and less space was available on the substrate [46]. The modelled summer Chl-a concentrations (proxy for phytoplankton bio-mass) were highest inside Horsens Fjord and decreased towards the more open waters (Fig 4B). The mussel farm significantly reduced Chl-a concentrations with up to 50% within the farm and showed reduced (>5%) values up to 5 km away from the farm (Fig 4C).

## Mussel farm

The 3D mussel farm model consists of a mussel population growth model embedded in a sim-ple, offline hydrodynamic model using FlexSem. The model uses prescribed velocity profiles (m/s) either obtained from a full 3D hydrodynamic model or monitoring data (e.g. ADCP data) and hence both current direction and speed change over time within the model domain. Further, vertical and horizontal diffusion is specified in the offline model: Horizontal diffusion as a constant and vertical diffusion as a function of wind speed. The model domain covers a rectangle of 1500x1050 m with a cell size of 50x50 m and 50x10m in the middle section (Fig 5A). The mussel farm is located in the middle and consists of 25 long-lines distributed with 10 m distance (250 m in total) each with a length of 200 m (50 m resolution) and vertical hanging rope-loops at 0–3 m depth (Fig 5B). Water column depth is 5 m with 1 m vertical resolution. From the open boundaries, temperature, salinity and phytoplankton biomass (expressed as Chl-a concentration) are transported into the area, but there are no internal processes such as phytoplankton growth, nutrient cycling, ect., which can be neglected due to the short residence time. The model runs during a typical farm production period from mussel larvae settling on the substrate in July until harvest in November or spring. The population growth model con-sists of an individual DEB model combined with the number of individuals per m of rope within the farm [43, 45]. The number of mussels decreases as the mussels get larger due to space restrictions, whereas the total mussel biomass in the farm increases over time. Mussel fil-tration within the farm removes phytoplankton biomass and the produced mussel feces are deposited in the sediment. The model can estimate mussel growth and biomass within the farm at high spatial resolution. In a model setup at Sallingsund in Limfjorden, results show that mussels reached a larger shell length in the upper left and lower right corners of the farm (Fig 5D) due to less food depletion and hence higher growth rate (Fig 5C). Since the model requires little forcing data, it can easily be set up for any area taking different environmental conditions into account for mussel growth. Forcing data for the Sallingsund setup was down-loaded from the ODA database.

## Sediment transport in the Clarion-Clipperton Zone

As part of the EU Horizon 2020 project Blue Nodules, a hydrodynamic model coupled with a sediment transport model was set up for an area in the Clarion-Clipperton Zone in the Pacific Ocean. The purpose of this model was to assess the distribution, fate and ecological impact of

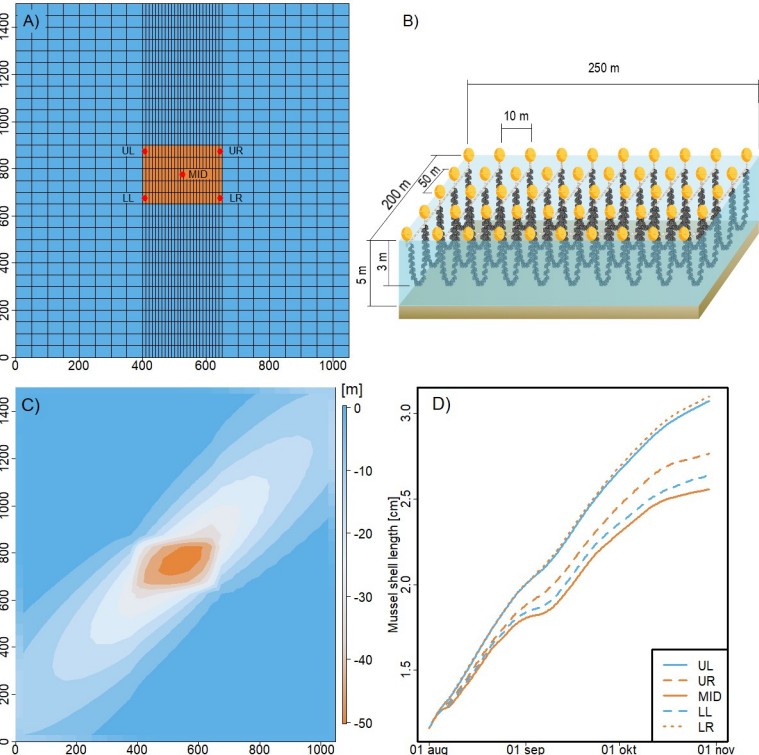

**Fig 5. Farm scale model.** A) Model grid with the mussel farm (250x200m) located in the middle (red cells) and 5 positions indicated: upper right (UR), upper left (UL), middle (MID), lower left (LL) and lower right (LR), B) conceptual diagram of the mussel farm and C) % Chl-a depletion in the farm area and D) increase of mussel shell length (cm) over time from the 5 positions in the farm.

far field sediment plumes originating from future nodule harvesting activities in the deep Pacific Ocean.

The computational mesh centered at (14.71N,125.44E) covered an area of approximately 150x150 km with a horizontal resolution varying from 2.9 km near the open boundary perimeter of the domain to 350 m in the central parts (Fig 6A). The water depth in the area varies between 4300m and 4700m and a model bathymetry was constructed by combining on-site multi-beam measurements (data courtesy of Global Sea Mineral Resources NV), and the 30 arc-second Smith and Sandwell bathymetry [47]. The model has 40 vertical layers, which varies in thickness from 500 m to 10 m around the abyssal depths were the sediment was released. Along the open boundary circumventing the model domain, the model was forced by velocities, water level, temperature and salinity provided by CMEMS. Sediment fractions and grain sizes were obtained using data from box cores collected on site by Global Sea Mineral Resources NV (GSR 2018). The sediment is very fine grained with a D50 of 9 μm. The total input and vertical distribution of sediment was provided by a near field, high-resolution CFD (computational fluid dynamics) model developed by project partner Royal IHC, The Netherlands.

The far field sediment model implements measured fall velocities for different grain size fractions and simulates a period of 30 days. Different mining scenarios including varying periods and patterns of nodule harvesting were considered, including a continuous mining activity over the full simulation period within a small area in the central part of the domain. The results demonstrate that the coarsest sediment fraction (66 μm) settles within several kilometers of

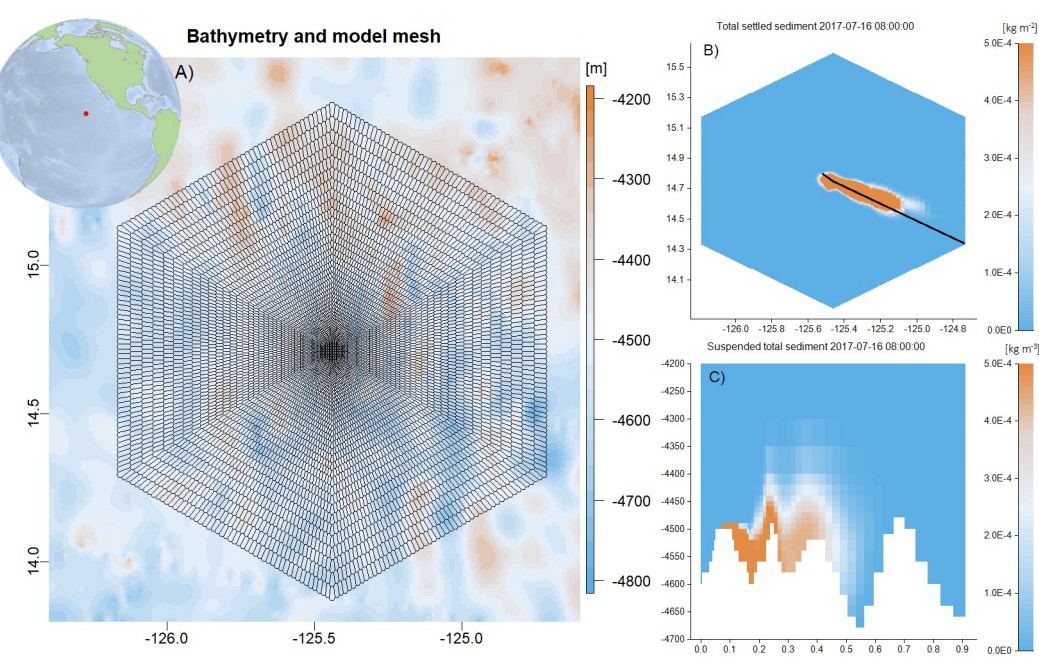

**Fig 6. Clarion-Clipperton Zone in the Pacific.** A) Location of model domain, computational mesh and bathymetry. B) Total sediment concentration (all fractions, kg m$^{-2}$) that has settled out on the seabed after 16 days of model simulation. C) suspended total sediment in the water column (kg m$^{-3}$) along a transect indicated by the black line B).

the mining site. Although bottom current velocities in the area typically are less than 10 cm/s, the finest sediment fractions are transported over much larger distances (Fig 6A and 6B).

## Zanzibar ABM

Unguja, also called "Zanzibar island" is located in the tropical Western Indian Ocean adjacent to the East African coast [48]. Zanzibar has been reported as a highly diverse seagrass area [49], with between 8 to 14 reported species. Seagrasses play an important role in shallow water ecosystems, enhancing biodiversity, acting as nursery areas, and promoting the nutrient cycle [50]. In this study, we developed an agent based model for Zanzibar island (Fig 7A). The computational mesh centered at (6.10575S,39.374W) has 6754 elements, 32 vertical layers and a total of 35,300 computational cells. The horizontal resolution varies from 250 meter around Stonetown west of Zanzibar to 3 km near the open boundaries and the vertical resolution varies from 3 meter in the upper layers to 100 meters in the deepest layers. A bathymetry was constructed by combing the GEBCO 15 arc second global bathymetry available at https://www.gebco.net/ with a high-resolution bathymetry for shallow areas created from satellite data by DHI GRASS. On the open boundaries the model was forced with temperature, salinity, water level and velocities obtained from CMEMS. Tide components obtained from the OCETide model (http://people.oregonstate.edu/~erofeevs/Afr.html) was added to the 24 hours CMEMS means of water level and velocities. The model includes a surface radiation model, which was forced by atmospheric wind velocities, air temperature, dew-point temperature and cloud cover downloaded from C3S. The timestep in the model was 4 minutes and the model was run for the entire 2018, although for this study it was run in August to match the flowering season of eelgrass.

The connectivity of the system was calculated based on the downstream connectivity probability, which quantifies the probability of the simulated shoots to end up in a particular

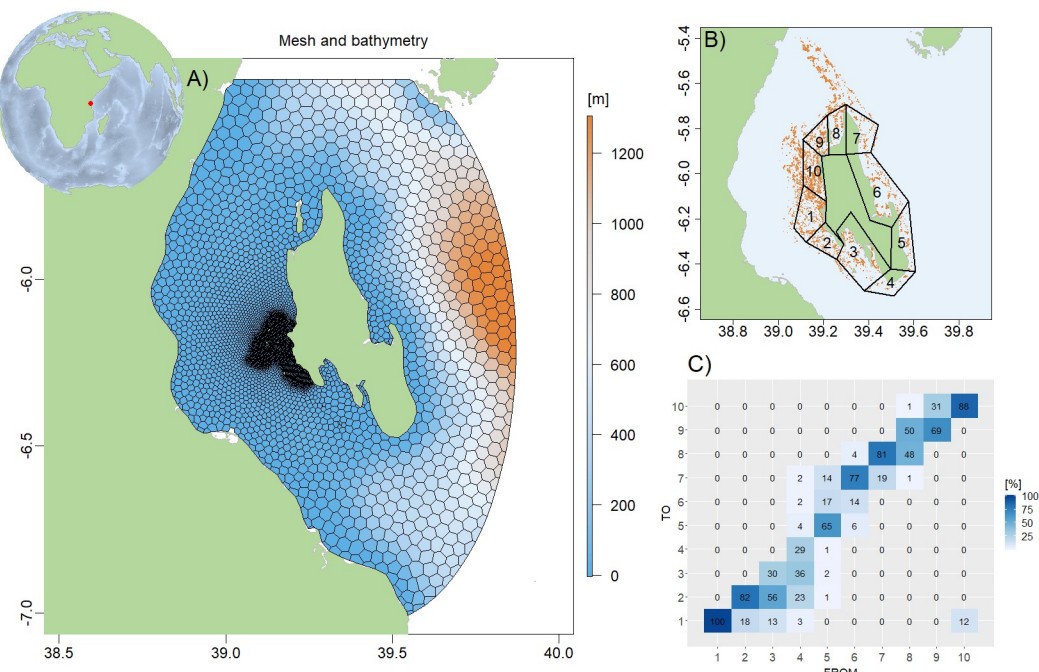

**Fig 7. Zanzibar, Tanzania, East Africa.** A) Computational mesh and bathymetry. B) Model area showing the agents (orange) and the area division into polygons. C) Downstream connectivity matrix for the 10 polygons in Zanzibar coastal area. The color code indicates the probability of the shoots from each polygon (FROM) to end up in each of the other 10 polygons (TO), with the diagonal elements being the probability of the shoots to stay in the same polygon (self-recruitment).

polygon from 1 to 10 (Fig 7B). A downstream connectivity matrix was then obtained from the output files (Fig 7C). The x and y-axis represents the source and sink polygons (from 1–10) respectively. The values in the matrix is the probability of the shoots in each polygon (1 to 10) to end up in another polygon (1 to 10). Observe that the self-recruitment (diagonal values in the matrix) is high in the North and Eastern parts of the island. In other words, most of the shoots released from the polygons (1–10) remain in the same polygon and are not dispersed to neighboring areas. From the model output, we can observe that currents are mainly to the NW. This explains the high transport of shoots from the western polygons to the adjacent areas.

## Discussion

The usage examples presented in this paper covers both ocean physics, marine biology, sediment transport and agent based modelling, which are the target topics of the currently implemented modules. The examples given are all three dimensional models, but the framework has been used with models from zero to 3 dimensions. Zero or one dimensional setups have proven particularly valuable in teaching and for ultra fast box models allowing automated optimization of calibration parameters where thousands of models runs are needed. Simpler very fast models can also be used in systems that implement ensemble data assimilation or in ecosystem management.

Even though creating a model and running it can be done without knowledge of c++ or fortran programming, it is still very useful to have some experiences in treating and visualizing data. When the model has run, the user would need to examine the output, which is best done in a high level programming language such as R or MATLAB. Simple models can be made by

using any text editor (see example in Appendix 1), but models that simulate an actual marine area, requires that the user can construct an orthogonal mesh for the area. While this can be done by using a mesh generator such as JigSaw, it still requires some programming experience. Mesh generators that are more commercial, but requires less programming experience, are also available.

The design that allows the user to input an equation system, which is then parsed and solved on the fly by a precompiled program, has some computational overhead. How much overhead, depends very much on the setup. When measuring the time it takes to solve identical equation system in the equation solver against compiled code, it is 36% faster in compiled code. But because the solving of the equations usually is a very small part of the computational effort in a typical model setup, where the bulk of the effort is spend on advection-diffusion, hydrodynamics and input/output, in a more realistic model setup, the overhead is much smaller. In the Zanzibar setup with a simple system of three pelagic equations and three variables without advection-diffusion, the overhead is 3% of the total simulation time. It would be possible for an experienced programmer to implement a compiled equation system in the framework, but the authors find the flexibility and ease of use more valuable than the small computational overhead.

The code that solves the advection-diffusion (AD) equation, the code that solves the user defined equations and parts of the hydrodynamic code are parallelized with OpenMP. This scales very well for e.g. AD when solving for many state parameters as a typical ecological equations system will contain. This is because the AD computations are independent for the different state parameters and therefore the threads that handle each state parameter do not need to communicate. However, the solution to the Navier-Stokes equation for the hydrodynamics is more complex. Here, the main computational effort goes into solving a system of linear equations using the preconditioned conjugate gradient method, a problem that is not well suited for OpenMP parallelization. Future development includes improving the parallelization of the hydrodynamic solver by utilizing Graphics Processing Unit (GPU) [51] or Message Passing Interface (MPI) [52].

Development of numerical models are an ongoing process as the numerical method and programming possibilities are constantly expanding and improving. The future development of FlexSem will focus on further improving the user friendliness, expanding the possible uses and implementing known and well-tested improvements on the numerical methods, parallelization and code optimization. Currently support for wetting and drying to simulate very shallow areas and/or areas with large tidal water level changes are being tested and is expected to be available in 2020. Waves can be very important when modelling sediment transport. There are no current plans to add a wave model module to the system. However, the user can include the effect of waves on sediment transport by implementing a forcing that modifies the diffusion and/or resuspension threshold. For Arctic applications, handling of sea-ice is essential, and it is planned to implement this as a forcing, which will affect heat exchange, light penetration and surface stress. Also, a flux correcting advection scheme and more efficient parallelization of the hydrodynamic solver are much desired.

## Conclusions

The FlexSem marine modelling framework has proven to be a very versatile and user-friendly option for marine modelling in different applications and environmental settings from the Arctic to temperate and tropical waters. The fact that it can be used on a desktop pc in a windows environment and by users not accustomed to programming (fortran or c++) makes it well suited for idealized studies or studies of smaller areas, typically coastal or estuarine

locations. Further, the framework has been used by both Ph.D. and master students as well as in teaching of non-modellers during a summer course, which underlines the user-friendliness of the system. The many possible combinations of modules, e.g. hydrodynamics-sediment, hydrodynamics-ABM, hydrodynamics-ecological-ABM and the possibility for the user to define customized equation systems for the pelagic, benthic and agent based models enhances the versatility of the system. We have demonstrated this by presenting examples of applications that are very different in both choice of modules, scales, location and purpose. The source code is freely available under the GNU license and this provides both the transparency and possibility for extending the code that are needed for scientific applications. The framework has a small computational overhead due to the approach of separating programming from modelling. Considering this and the fact that the parallelization can be improved makes the system best suited for smaller model domain and idealized setups.

## Supporting information

**S1 File. FlexSem setup example.**
(DOCX)

**S1 Fig. Model output.**
(PNG)

## Acknowledgments

We acknowledge the Danish Meteorological Institute (DMI), the Geological Survey of Denmark and Greenland (GEUS), Global Sea Mineral Resources (GSR) and the Copernicus Marine Environment Monitoring Service (CMEMS) and DHI GRASS for providing forcing data for the model setups.

## Author Contributions

**Conceptualization:** Janus Larsen, Christian Mohn, Ane Pastor, Marie Maar.

**Formal analysis:** Janus Larsen, Christian Mohn, Ane Pastor, Marie Maar.

**Funding acquisition:** Christian Mohn, Marie Maar.

**Methodology:** Janus Larsen, Christian Mohn, Ane Pastor, Marie Maar.

**Software:** Janus Larsen.

**Visualization:** Janus Larsen.

**Writing – original draft:** Janus Larsen, Christian Mohn, Ane Pastor, Marie Maar.

**Writing – review & editing:** Janus Larsen, Christian Mohn, Ane Pastor, Marie Maar.

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
