## [Decision Letter · Decision Letter 0]

5 Mar 2020

PONE-D-20-01866

A versatile marine modelling tool applied to arctic, temperate and tropical waters

PLOS ONE

Dear Janus and co-authors,

Thank you for submitting your manuscript to PLOS ONE. After careful consideration, we feel that it has merit but does not fully meet PLOS ONE’s publication criteria as it currently stands. Based upon the evaluation of your manuscript by two expert reviewers and my own reading my decission is "Minor revision". Therefore, we invite you to submit a revised version of the manuscript that addresses the (mainly minor) issues pointed to by the referees.

In particular

a) please look at what Rev 1 writes about data. I don't see any real issue here. Rev 2 writes that the only data you need to be able to set up the framework is some forcing data. Still, you may need to rewrite a little on data availability.

b) please also consider this requirement I've got from Associate Editor Miquel Vall-llosera Camps (mvall-lloseracamps@plos.org): In your evaluation of this manuscript, please consider in addition whether it meets PLOS ONE criteria for manuscripts that describe new methods. Specifically, these reports must meet the criteria of utility, validation, and availability, which are described in detail at http://journals.plos.org/plosone/s/submission-guidelines#loc-methods-software-databases-and-tools.

Again, I don't see any problems here. As Rev 2 points out, you have made the code open and freely distributed via a web page within the GNU General Public License.

We would appreciate receiving your revised manuscript by Apr 19 2020 11:59PM. To enhance the reproducibility of your results, we recommend that if applicable you deposit your laboratory protocols in protocols.io, where a protocol can be assigned its own identifier (DOI) such that it can be cited independently in the future. For instructions see: http://journals.plos.org/plosone/s/submission-guidelines#loc-laboratory-protocols

We look forward to receiving your revised manuscript.

Kind regards,

Geir Ottersen

Academic Editor

PLOS ONE

Journal Requirements:

2. In your Methods section, please provide additional location information of the study areas, including geographic coordinates for the data set if available.

3. Please include your tables as part of your main manuscript and remove the individual files. Please note that supplementary tables (should remain/ be uploaded) as separate "supporting information" files.

Reviewers' comments:

Reviewer's Responses to Questions

**Comments to the Author**

1. Is the manuscript technically sound, and do the data support the conclusions?

Reviewer #1: Yes

Reviewer #2: Yes

2. Has the statistical analysis been performed appropriately and rigorously? 

Reviewer #1: Yes

Reviewer #2: I Don't Know

3. Have the authors made all data underlying the findings in their manuscript fully available?

Reviewer #1: No

Reviewer #2: Yes

4. Is the manuscript presented in an intelligible fashion and written in standard English?

Reviewer #1: Yes

Reviewer #2: Yes

5. Review Comments to the Author

Reviewer #1: General comments:

The work presented a versatile community framework for 3D hydrodynamic, biogeochemical, individual based and sediment transport modelling with flexible grids, which enable researchers to conduct the coupled 3D environmental and biogeochemical modelling by providing a pre-compiled portable framework in which users can pick any combinations of the models. The physics and numerical solutions of the problem have been properly addressed and 5 model applications are demonstrated.

The objective of the FlexSem model framework is to make advanced 3D unstructured marine modelling more available for a broad range of applications and users, offering non-specialists and early stage researchers easier access to ecosystem modelling without the need of expert knowledge in computing programming. In Europe there are not many similar synthesized community modelling frameworks for users, especially for coastal-estuary applications.

1. Authors stated that the model system is “user friendly”. From the text, I can see some features showing the easy-use of the model. However, by looking at the model website, there is only a 0D box model setup. For other test cases and setups, only general information is given but lack of detailed setups. I wonder if the model has been used by a group of users that the model is “user friendly”.

2. On the data availability, PLOS One asked authors to make all relevant data available for open access, this is only done partly.

3. There are still some emerging issues: 1) for applications in coastal-estuary continuum, flooding and drying and heat flux from sea bottom are important issues; 2) for Arctic applications, ice modelling is essential; 3) for sediment modelling, waves are essential. The corresponding treatments in the model, if not perfect or not available, can be mentioned in the discussion for future improvement.

4. Acronyms should be mentioned as full name when used first time, e.g., AD: advection-deffusion, INTAROS etc. I suggest that authors to have a careful grammar check on the text.

Specific comments:

P4: Change “can be setup” to “can be set up”

P5: Change “enables the user create and test model parts” to “enables the user to create and test model parts”

P6: Change “by solving an equations system” to “by solving a set of equations”

P12: “The modelled salinity and salinity profiles” needs to be changed. Fig 3B and 3C show validation for two salinity profiles which are very similar. Since it is a CTD measurement, Fig 3B should be changed to a temperature validation.

P12: change “Disko bay” to “Disko Bay”

P11-12: treatment of ice should be mentioned in the Arctic application

P17: Change “some experience” to “some experiences”

p19: “arctic” should be “Arctic”

Reviewer #2: GENERAL COMMENTS:

The paper would go within the catagory of "Submissions describing methods, software, databases, or other tools".

The manuscript describes a model system aimed to be used within marine environments, especially 3D hydrodynamics, biogeochemical, individual based and sediment transport modelling. The system is explained to be user friendly and one need not to be an expert in programming to be able to use it. The model framework is aimed to be used by scientists, but can also be used in school environments. There are five examples of applications for the model system, briefly described in the manuscript. The code is open and freely distributed via a web page within the GNU General Public License. The only data you need to be able to set up the framework is some forcing data.

The manuscript is very well written, however, it is very much a descriptive folder for a product. And a thoroughly discussion about validation, weaknesses and limitations is missing.

SPECIFIC COMMENTS:

1. The velocity arrows show at each intersection only one direction. Does this mean that the transports only go one direction, or can the velocity be also negative?

2. A citation to the study in Western Greenland is missing, where more information about the model setup, validation, results and discussion can be found.

3. On page 13, it is not clear that the mussel farm reduced Chl-a concentration with up to 50 %. It seems more like the change is zero in the middle of the farm. This is probably just bad choice of colour. Please change the colour palette. However, any 50 % (orange colours) can not be seen in the figure. From where do you get this number?

4. Also, I would prefer to have the units of the parameters right at the colour bars in all the figures where it is missing. It makes the figures easier to understand.

5. A citation to the study in Horsens Fjord area is missing, where more information about the model setup, validation, results and discussion can be found.

6. A citation to the mussel farm study would be nice, where more detailed information about the model setup, validation, results and discussion can be found.

7. A citation to the sediment transport study would be nice, where more detailed information about the model setup, validation, results and discussion can be found.

8. The sentence describing the connectivity matrix is hard to understand, and should therefore be re-formulated. Or extended for easier understanding. And also, which number corresponds to which polygon?

9. The discussion is not complete and a paragraph about limitations and weaknesses should be included.

10. A discussion about the evaluation of model results is also missing. How accurate is the model system? How do the user know what to evaluate and how?

11. In the discussion on page 17, what do you mean with zero-dimension? In the framework description 1D, 2D and 3D are mentioned, but not zero D.

MINOR EDITS:

12. Change “show” to “shown” at page 7, 4th paragraph. “It can be shown…. ”

13. A “.” is missing at page 12, after the citation [41].

14. On page 12, there is a citation to fig. 4A when talking about the position of the mussel farm. However, the mussel farm is not shown in Fig. 4A.

15. In page 16, there is a citation missing to fig. 7B, since the citation to fig. 7B should be 7C.

16. Remove the “In” in the beginning of the discussion, page 17. Remove also the comma after “paper”.

17. In discussion, page 17, second paragraph, change “ … output, and this is….” to “… output, which is ….”

18. Make sure the references are uniform. Specifically change reference nr. 32 to lowercase.

19. Fig.1 is a bit blurred.

20. Please make clear in the figure text for fig. 2 that HD means Hydrodynamics.

21. The font size in figure 5b is way to small. Impossible to read. Please make larger.

6. PLOS authors have the option to publish the peer review history of their article (what does this mean?). If published, this will include your full peer review and any attached files.

Reviewer #1: Yes: Jun She

Reviewer #2: No

---

## [Author Response · Author response to Decision Letter 0]

14 Mar 2020

Dear Editor and reviewers,

Please find our responses to your comments and questions below.

Best regards,

Janus Larsen

Department of BioScience, Aarhus University, Denmark

janus@bios.au.dk

Editor

We have verified that the manuscript meets PLOS ONE's style requirements, inserted the table in the manuscript, added geographical coordinates for the study areas, put the supporting information in a separate file and included supporting information headers at the end of the manuscript.

a) Please see our answer to reviewer 2’s question on data availability.

b) The presented modelling software FlexSem meets the publication criteria for utility, validation and availability because the precompiled program and source code is freely available at the website within the GNU General Public License. Furthermore it is, to our knowledge, the only free and open source marine modelling software available, that offers the descripted range off application and ease of use through a precompiled program were models can be set up without formalized programming or compiling.

Reviewer 1

1. Authors stated that the model system is “user friendly”. From the text, I can see some features showing the easy-use of the model. However, by looking at the model website, there is only a 0D box model setup. For other test cases and setups, only general information is given but lack of detailed setups. I wonder if the model has been used by a group of users that the model is “user friendly”.

The user friendliness arises mainly from the fact that the system can be used to make dynamic models using just a text editor and the precompiled executable. No expert knowledge on object orientated programming, parallelization, compiler options or Unix skills are needed. We have chosen to make a simple setup that demonstrates all the basic functions available at the website. More advanced users must refer to the documentation also available at the website. The idea is that the users can download and use the precompiled executable and available functions to make their own setups for a given area following the simple example and documentation on the web page. This framework has been tested on a several master and PhD students with success. The case studies in the paper are demonstrations of how the model framework can be applied, but it is up to the users to define their own setups.

2. On the data availability, PLOS One asked authors to make all relevant data available for open access, this is only done partly.

The manuscript has been updated with more information about were forcing data can be found. The majority of the data can be downloaded directly from the web (CMEMS, ODA, PROMICE, GEBCO, SMITH and SANDWELL) and here we now provide links to where the data can be found or refer to the relevant publications. In a few cases data is available on request (e.g. sediment data from EU project Blue Nodules), in which case the data owner is listed.

3. There are still some emerging issues: 1) for applications in coastal-estuary continuum, flooding and drying and heat flux from sea bottom are important issues; 2) for Arctic applications, ice modelling is essential; 3) for sediment modelling, waves are essential. The corresponding treatments in the model, if not perfect or not available, can be mentioned in the discussion for future improvement.

We thank the reviewer for pointing out these important shortcomings and have extended the discussion sections on limitations and future work to include these points. How absorption and reflection of heat at the sea bottom in shallow water is treated is described in the section on surface heat model.

4. Acronyms should be mentioned as full name when used first time, e.g., AD: advection-deffusion, INTAROS etc. I suggest that authors to have a careful grammar check on the text.

The manuscript has been updated so that abbreviations always are written out the first time they are used.

Specific comments:

P4: Change “can be setup” to “can be set up”

Done.

P5: Change “enables the user create and test model parts” to “enables the user to create and test model parts”

Done.

P6: Change “by solving an equations system” to “by solving a set of equations”

Done.

P12: “The modelled salinity and salinity profiles” needs to be changed. Fig 3B and 3C show validation for two salinity profiles which are very similar. Since it is a CTD measurement, Fig 3B should be changed to a temperature validation.

Fig. 3C has been updated to show a salinity profile and the corresponding temperature profile.

P12: change “Disko bay” to “Disko Bay”

Done.

P11-12: treatment of ice should be mentioned in the Arctic application

This is now mentioned in the discussion.

P17: Change “some experience” to “some experiences”

Done.

p19: “arctic” should be “Arctic”

Done.

Reviewer 2

SPECIFIC COMMENTS:

1. The velocity arrows show at each intersection only one direction. Does this mean that the transports only go one direction, or can the velocity be also negative?

It has now been clarified in the text that the velocities shown in fig 1 can also be negative. This is also described in detail in reference 26 [Casulli & Zanolli 2002].

2. A citation to the study in Western Greenland is missing, where more information about the model setup, validation, results and discussion can be found.

The text has been updated and now states that more information on this setup can be found at the FlexSem website. Furthermore, one or several publications on the Disko Bay modelling are planned as part of the INTAROS project, which will be announced at the website when available.

3. On page 13, it is not clear that the mussel farm reduced Chl-a concentration with up to 50 %. It seems more like the change is zero in the middle of the farm. This is probably just bad choice of colour. Please change the colour palette. However, any 50 % (orange colours) can not be seen in the figure. From where do you get this number?

The figure has been updated so that the position of the musselfarm now is shown in Fig. 4A instead of Fig. 4C. The 50% reduction is now more visible in Fig. 4C 

4. Also, I would prefer to have the units of the parameters right at the colour bars in all the figures where it is missing. It makes the figures easier to understand.

All color-coded plots have been updated with the color legend labeled with the unit. 

5. A citation to the study in Horsens Fjord area is missing, where more information about the model setup, validation, results and discussion can be found.

This work has not been published yet. The final report of the BONUS OPTIMUS project, which will be published in 2020 contains more information. Also, a publication on the Horsens Fjord modelling is planned as part of the OPTIMUS project and will be announced at the FlexSem website when available.

6. A citation to the mussel farm study would be nice, where more detailed information about the model setup, validation, results and discussion can be found.

This work has not been published yet. However a manuscript has been submitted (von Thenen, M., Maar, M., Hansen, H.S., Friedland, R., Schiele, K.S. Applying a combined geospatial and ecological model to identify suitable locations for mussel farming. Marine Pollution Bulletin.) and it will be announced at the FlexSem website when available.

 7. A citation to the sediment transport study would be nice, where more detailed information about the model setup, validation, results and discussion can be found.

This work has not been published yet. The final report of the EU Blue Nodules, which will be published in 2020 contains more information. Also, a publication focused on the sediment modelling in the CCZ are in working progress. It will be announced at the FlexSem website when available.

8. The sentence describing the connectivity matrix is hard to understand, and should therefore be re-formulated. Or extended for easier understanding. And also, which number corresponds to which polygon?

The explanation of the connectivity matrix has been rephrased and Fig.4B has been updated with the polygon numbers.

9. The discussion is not complete and a paragraph about limitations and weaknesses should be included.

The discussion has been extended with a section on limitations and weaknesses.

10. A discussion about the evaluation of model results is also missing. How accurate is the model system? How do the user know what to evaluate and how?

The case studies in the paper are demonstrations of how the model framework can be applied and it is out of scope for this paper to show validations for all studies. Further, it is up to the users to make their own model setups and to validate their models. There exist a large literature on how to evaluate models. Publications with model validation results for all the usage examples are in progress or planned and will be will be announced at the FlexSem website when available.

11. In the discussion on page 17, what do you mean with zero-dimension? In the framework description 1D, 2D and 3D are mentioned, but not zero D.

This has now been clarified in the framework description.

MINOR EDITS:

12. Change “show” to “shown” at page 7, 4th paragraph. “It can be shown…. ”

Done.

13. A “.” is missing at page 12, after the citation [41].

Done.

14. On page 12, there is a citation to fig. 4A when talking about the position of the mussel farm. However, the mussel farm is not shown in Fig. 4A.

Figure 4A has been updated to show the location of the farm.

15. In page 16, there is a citation missing to fig. 7B, since the citation to fig. 7B should be 7C.

Corrected and a reference to Fig. 7B added.

16. Remove the “In” in the beginning of the discussion, page 17. Remove also the comma after “paper”.

Done.

17. In discussion, page 17, second paragraph, change “ … output, and this is….” to “… output, which is ….”

Done.

18. Make sure the references are uniform. Specifically change reference nr. 32 to lowercase.

Done.

19. Fig.1 is a bit blurred.

A new version higher resolution version of fig 1 has now been provided.

20. Please make clear in the figure text for fig. 2 that HD means Hydrodynamics.

The figure text has been updated.

21. The font size in figure 5b is way to small. Impossible to read. Please make larger.

The font size has been increased.

---

## [Editor Report · Decision Letter 1]

19 Mar 2020

A versatile marine modelling tool applied to arctic, temperate and tropical waters

PONE-D-20-01866R1

DearJanus and co-authors,

We are pleased to inform you that your manuscript has been judged scientifically suitable for publication and will be formally accepted for publication once it complies with all outstanding technical requirements. You have done a good and efficient job in adressing the reviewers' requests, as far as I can see they have all been dealt with satisfactorily. I therefore see no remaining scientific issues, other editors will deal with technical issues, as described in the following.

With kind regards,

Geir Ottersen

Academic Editor

PLOS ONE

---

## [Editor Report · Acceptance letter]

24 Mar 2020

PONE-D-20-01866R1 

A versatile marine modelling tool applied to arctic, temperate and tropical waters 

Dear Dr. Larsen:

I am pleased to inform you that your manuscript has been deemed suitable for publication in PLOS ONE. Congratulations! Your manuscript is now with our production department. 

With kind regards,

on behalf of

Dr. Geir Ottersen 

Academic Editor

PLOS ONE